# Development of an antigen detection assay for early point-of-care diagnosis of *Zaire ebolavirus*

**Haley L. DeMers**[1], **Shihua He**[2], **Sujata G. Pandit**[1], **Emily E. Hannah**[1], **Zirui Zhang**[2,3], **Feihu Yan**[3], **Heather R. Green**[1], **Denise F. Reyes**[1], **Derrick Hau**[1], **Megan E. McLarty**[1], **Louis Altamura**[4¤], **Cheryl Taylor-Howell**[4], **Marcellene A. Gates-Hollingsworth**[1], **Xiangguo Qiu**[2]*, **David P. AuCoin**[1]*

**1** Department of Microbiology and Immunology, University of Nevada, Reno School of Medicine Reno, Nevada, United States of America, **2** Special Pathogens Program, National Microbiology Laboratory, Public Health Agency of Canada, Winnipeg, Manitoba, Canada, **3** Department of Medical Microbiology, University of Manitoba, Winnipeg, Manitoba, Canada, **4** Diagnostic Systems Division, United States Army Medical Research Institute of Infectious Diseases, Fort Detrick, Maryland, United States of America

¤ Current address: National Biodefense Analysis and Countermeasures Center, Operated by Battelle National Biodefense Institute for the US Department of Homeland Security, Frederick, Maryland, United States of America

* xiangguo.qiu@canada.ca (XQ); daucoin@med.unr.edu (DPA)

**Data Availability Statement:** All relevant data are within the manuscript and its Supporting Information files.

## Abstract

The 2013–2016 Ebola virus (EBOV) outbreak in West Africa and the ongoing cases in the Democratic Republic of the Congo have spurred development of a number of medical countermeasures, including rapid Ebola diagnostic tests. The likelihood of transmission increases as the disease progresses due to increasing viral load and potential for contact with others. Early diagnosis of EBOV is essential for halting spread of the disease. Polymerase chain reaction assays are the gold standard for diagnosing Ebola virus disease (EVD), however, they rely on infrastructure and trained personnel that are not available in most resource-limited settings. Rapid diagnostic tests that are capable of detecting virus with reliable sensitivity need to be made available for use in austere environments where laboratory testing is not feasible. The goal of this study was to produce candidate lateral flow immunoassay (LFI) prototypes specific to the EBOV glycoprotein and viral matrix protein, both targets known to be present during EVD. The LFI platform utilizes antibody-based technology to capture and detect targets and is well suited to the needs of EVD diagnosis as it can be performed at the point-of-care, requires no cold chain, provides results in less than twenty minutes and is low cost. Monoclonal antibodies were isolated, characterized and evaluated in the LFI platform. Top performing LFI prototypes were selected, further optimized and confirmed for sensitivity with cultured live EBOV and clinical samples from infected non-human primates. Comparison with a commercially available EBOV rapid diagnostic test that received emergency use approval demonstrates that the glycoprotein-specific LFI developed as a part of this study has improved sensitivity. The outcome of this work presents a diagnostic prototype with the potential to enable earlier diagnosis of EVD in clinical settings and provide healthcare workers with a vital tool for reducing the spread of disease during an outbreak.

**Funding:** This study was supported by Department of Homeland Security contract HSHQDC-15-C-B0067 (DPA and XQ) (https://www.dhs.gov/) and the Public Health Agency of Canada (XQ) (http://www.phac-aspc.gc.ca/). Following completion of this research, the project was funded through NIAID (STTR 1R41AI149940). InBios International Inc. (Seattle WA) was awarded the contract with a subcontract issued to the AuCoin laboratory at the University of Nevada, Reno. The funders had no role in study design, data collection and analysis, decision to publish, or preparation of the manuscript.

**Competing interests:** I have read the journal's policy and the authors of this manuscript have the following competing interests: InBios International Inc. (Seattle WA) was awarded the contract with a subcontract issued to the AuCoin laboratory at the University of Nevada, Reno. The authors have declared that no competing interests exist.

## Author summary

Ebola virus (EBOV) causes a severe hemorrhagic fever and has an extremely high fatality rate that ranges from 60%-90%. There is no approved treatment or vaccine for this infectious disease and halting spread of the virus relies on identifying and isolating infected patients quickly. The current gold standard, polymerase chain reaction assay, requires patient samples be transported to regional reference laboratories where it often takes days to get results. A handful of Ebola rapid diagnostic tests have been developed, but lack the sensitivity required to detect the virus in earlier stages of the disease. There is great need for more sensitive rapid diagnostic tests that can identify the EBOV infected patients when they first become symptomatic. This study focused on production of high affinity mAbs to two target EBOV proteins for development of a more sensitivity rapid diagnostic test. Efforts have resulted in production of prototype detecting the EBOV glycoprotein that shows a notable improvement in sensitivity and offers the potential for earlier diagnosis of infection.

## Introduction

The West African Ebola outbreak that occurred from 2013–2016, caused by Ebola virus (EBOV; Species: *Zaire ebolavirus*), was the most severe outbreak in the history of the disease. The outbreak had more cases, deaths and survivors than all other outbreaks combined; 28,652 cases, 11,325 deaths, 17,300 estimated survivors [1–3]. Additionally, the second largest outbreak is currently taking place in the Democratic Republic of the Congo, with 3392 confirmed and probable cases as of January 2020 [4]. A primary challenge in managing EBOV outbreaks is the ability to quickly and accurately identify which patients presenting with illness near the outbreak are in fact Ebola virus disease (EVD) patients [5, 6]. During the 2013–2016 outbreak, point-of-care (POC) diagnosis of EBOV relied heavily on diagnosis of symptoms by case definition, a difficult task given the non-specific symptoms that present early in EVD patients [5, 7]. The gold standard diagnostic tests, polymerase chain reaction (PCR) assays, are lab-based assays that require samples be collected and stored until healthcare workers arrange transport to centralized laboratory facilities. These tests are not routinely conducted at the POC due to requirements that include significant infrastructure and skilled laboratory personnel [8]. It was noted after the West African outbreak, that even with the deployment of mobile laboratories, the delay from sample acquisition to patient result was 5 days on average and significantly longer during bursts of cases where samples overwhelmed laboratory capabilities [9].

The lateral flow immunoassay (LFI) test, a platform that uses antibody-based technology integrated into a membrane-based assay, has several strengths that are well suited to fill the need for a rapid EBOV diagnostic. The LFI can be performed at the POC without the need for sophisticated laboratory equipment or highly trained personnel. It is also affordable, easy to distribute, and most LFIs have a 18-month shelf life without requiring refrigeration. The LFIs biggest strength is that they can provide a result in less than twenty minutes [10]. With many strengths, there are also challenges that come with utilizing the LFI platform. There is an unavoidable dependence on two factors; the need for highly reactive antibodies to a disease target and sufficient concentrations of the target available for detection within clinical samples. For detection of disease during the early stages of infection where target antigens may be at lower concentrations, a greater reliance falls on the affinity or binding reactivity of the antibodies and their overall functionality in the assay. This is especially true for EBOV, where

virions and shed viral antigens may be in lesser quantities during the early stages of infection [11]. A successful EBOV LFI must employ high affinity antibodies to maximize sensitivity and improve chances of detecting the virus earlier in the course of disease.

There are seven genes that make up the EBOV genome. The envelope glycoprotein (GP) and the viral matrix protein (VP40) are both viral gene products that have been previously identified as targets in infected patient samples [12–15]. GP is displayed on the surface of the virion and is also present in a shed form during infection [14, 16]. VP40 plays an important role in assembly and budding, and is the most abundantly expressed filovirus protein [17, 18]. While it is known that the viral load can range from $5x10^1$ to $7x10^7$ pfu/mL in serum from EBOV infected patients, there is limited data on the concentration of GP and VP40 proteins in clinical samples [19–21].

The goal of this study was to isolate high affinity monoclonal antibodies (mAbs) reactive to EBOV GP and VP40, and integrate the mAbs into a LFI capable of early POC diagnosis of EVD. Thirty-two mAbs were produced by immunizing mice with either EBOV virus-like particles (eVLPs) or recombinant VP40 (rVP40). eVLPs are produced by expressing GP and VP40 within mammalian cells; the resulting particles mimic the morphology of live virus [22]. These particles lack the EBOV genome and are therefore noninfectious and can be used under lower biosafety precautions than the virus itself [23, 24]. Reactivity of each mAb was determined by enzyme-linked immunosorbent assay (ELISA), surface plasmon resonance (SPR) and Western immunoblot. Additionally, all pairwise combinations of mAbs were assessed in LFI format for selection of pairs with the greatest analytical sensitivity and lowest background signal. Selected mAb pairs were used to produce optimized LFIs that were evaluated for reactivity with live EBOV (variant Makona; from the 2013–2016 West Africa outbreak) present in tissue culture supernatant and sera collected from infected non-human primates (NHP). All testing with live EBOV was done in parallel with a commercially available EBOV RDT (ReEBOV Antigen Rapid Test, Zalgen Labs) for comparison.

## Methods

### eVLP and nano-eVLP production

eVLPs were produced as previously described by Wahl-Jensen et al [22]. Briefly, 293FT cells were co-transfected with plasmids expressing the EBOV Zaire glycoprotein (GenBank accession L11365; BEI Resources, NR-19814) and the EBOV Zaire matrix protein 40 fused with beta-lactamase (GenBank accession L11365; BEI Resources, NR-19813) using *Trans*IT-293 per manufacture's recommendations (Mirus, Madison, WI). After transfection, cells were incubated for 72 hours. Supernatant was clarified and purified over a 20% sucrose cushion. Nano-eVLPs were produced as previously described [25]. Briefly, eVLPs were isolated and then sonicated for nine pulses, one second each. This process was used to produce smaller "nano-eVLPs" that could be 0.2 μm filter sterilized prior to mouse immunizations, as eVLPs are not as easily filtered. eVLPs and nano-eVLPs were quantitated with BCA Assay (ThermoFisher Scientific, Grand Island, NY).

### mAb production

Female CD1 mice, 6 to 8 weeks-old (Charles River Laboratories, Inc., Frederick, MD), were immunized with EBOV nano-eVLPs or rVP40 (IBT Bioservices, Rockville, MD) mixed with Freund's Adjuvant (Sigma-Aldrich, St. Louis, MO) via intraperitoneal injection. An eVLP indirect ELISA (described below) was used to determine serum antibody EBOV titers at 9, 11, and 13 weeks post-immunization. Three days prior to spleen harvest, one final dose of nano-eVLP or rVP40 was administered via intravenous injection. Hybridoma cells were produced

using standard techniques [26]. Hybridoma supernatant was collected, and mAbs were purified using recombinant protein A affinity column chromatography.

### eVLP Indirect ELISA

Microtiter plates were coated with 100 μL/well of 0.005% poly-L-lysine (ThermoFisher Scientific, Grand Island, NY) for 90 minutes at 37˚C. Plates are then washed with PBS and coated with 2.5 μg/mL eVLP (100 μL/well) overnight at room temperature. The plates are washed with PBS containing 0.05% Tween 20 (PBS-T) and blocked for 90 minutes at 37˚C with blocking buffer (5% non-fat milk, 0.5% Tween 20 in PBS), followed by another washing step with PBS-T. The primary antibody was added to the first well and serial two-fold dilutions were performed across the plates; starting concentration of 1 μg/mL for purified mAb or 1:1000 dilution of mouse serum (100 μL/well). After a 60-minute incubation at room temperature, the plates were washed with blocking buffer and incubated with horseradish peroxidase (HRP) labeled goat anti-mouse IgG antibody (SouthernBiotech, Birmingham, AL) or IgG subclass-specific goat anti-mouse antibody (SouthernBiotech, Birmingham, AL) at 1:5000 dilution (100 μL/well). The plates were washed with PBS-T and incubated with tetramethylbenzidine substrate (100 μL/well) (Kirkegaard & Perry Laboratories, Inc., Gaithersburg, MD). The reaction was stopped after 30 minutes with 1M $H_3PO_4$ (100 μL/well). Plates were read at an optical density of 450nm ($OD_{450}$).

### Western immunoblot

A standard Western immunoblot procedure was done with semidry blotting. Samples, either 1 μg of His-tagged recombinant EBOV GP minus the transmembrane domain (rGP) (IBT Bioservices, Rockville, MD), 0.5 μg of rVP40 (IBT Bioservices, Rockville, MD) or 1 μg of eVLP, was mixed with 6x loading buffer, boiled and loaded onto the gel. Samples were separated on 10% SDS gel (Bio-Rad Laboratories, Hercules, CA) followed by transfer to nitrocellulose membrane (Bio-Rad Laboratories, Hercules, CA). GP and VP40 mAbs were used to probe the membrane as the primary antibody (200 ng/mL). HRP labeled goat anti-mouse IgG was used at a 1:10,000 dilution for detection. Signal was detected with a chemiluminescent substrate (ThermoFisher Scientific, Grand Island, NY) and imaged using a ChemiDoc XRS imaging system (Bio-Rad Laboratories, Hercules, CA).

### Antigen-capture ELISA

Microtiter plates were coated overnight with 100 μL/well of the capture mAb (1 μg/mL) in PBS. Plates were washed with PBS-T and blocked for 90 minutes at 37˚C with blocking buffer. eVLPs (10 μg/mL) were added to the first well and serial diluted two-fold across each plate in blocking buffer for a final volume of 100 μL/well. Plates were incubated for 60 minutes at room temperature, washed with PBS-T and incubated for another 60 minutes at room temperature with HRP-labeled detection mAb (1 μg/mL) diluted in blocking buffer (100 μL/well). HRP labeling of mAbs was performed using EZ-Link Plus Activated Peroxidase (ThermoFisher Scientific, Grand Island, NY). Plates were washed with PBS-T and incubated with tetramethylbenzidine substrate (100 μL/well). The reaction was stopped after 30 minutes with 1M $H_3PO_4$ (100 μL/well). Plates were read at an optical density of 450nm ($OD_{450}$). The LOD of the preliminary assays was defined as three-times background.

### Optimization of antigen-capture ELISA

Checkerboard ELISAs were performed to optimize the coating and detection mAb concentrations for the antigen-capture ELISA. Capture and detection mAbs were tested from 0.16–

10 µg/mL. The antigen-capture ELISA protocol, listed above, was used for incubation and washing steps. Final concentrations were chosen based on the lowest LOD of eVLP at three-times background signal. Final optimized concentrations for each mAb pair were: 1HK7 capture (5 µg/mL) and 1HK4 detection (0.625 µg/mL), 1HK7 capture (5 µg/mL) and 1HK11 detection (0.625 µg/mL), 2HK1 capture (5 µg/mL) and 2HK7 detection (1.25 µg/mL) and 2HK12 capture (5 µg/mL) and 2HK1 detection (0.625 µg/mL).

## Surface plasmon resonance (SPR)

All SPR experiments were performed using a Biacore X100 instrument (GE Healthcare, Piscataway, NJ). SPR studies to confirm binding of mAbs to rGP or rVP40 were done by capturing his-tagged antigen (rGP or rVP40) (10 µg/mL) onto the surface of a CM5 chip using a His Capture Kit (GE Healthcare, Piscataway, NJ). Binding was measured by a single injection (25 µg/mL) of purified antibody over the sensor chip. SPR experiments evaluating binding of mAbs to nano-eVLPs were performed by immobilizing nano-eVLPs onto the surface of a CM5 sensor chip; a reference flow was left blank for background subtraction (GE Healthcare, Piscataway, NJ). Qualitative analysis for initial screening of mAbs was performed by injection of a single concentration of purified antibody (25 µg/mL) over the sensor chip. Full kinetic analysis was done by injecting purified antibody over the sensor chip at concentrations ranging from 50–0.78 µg/mL in triplicate.

## Lateral flow immunoassay (LFI) screening

Initial LFI screening was performed with all GP or VP40 mAbs in the capture and detection position. A standard basic LFI design was used for broad screening and eVLPs spiked into chase buffer (1% casein in PBS) or chase buffer alone were evaluated. Briefly, 40 µL of sample was added to the conjugate pad, then the LFI was placed vertically in a microtiter well with 150 µL of chase buffer and allowed to run for 15–20 minutes. LFIs were visually evaluated and also read using a Qiagen ESE reader (Qiagen, Hilden, Germany). LFIs were ranked based on highest test line intensity (eVLPs spiked in buffer) minus background signal (chase buffer alone).

## GP-specific and VP40-specific LFI prototypes

The GP-specific LFI prototype was developed using mAb 1HK7 (capture) and mAb 1HK11 (detection) and the VP40-specific LFI prototype was developed with mAb 2HK1 (capture) and mAb 2HK12 (detection). Capture mAb was sprayed onto nitrocellulose membranes as the test line at 1 mg/mL using a BioDot XYZ platform (BioDot, Irvine, CA). Goat anti-mouse Ig (SouthernBiotech, Birmingham, AL) was sprayed as the control line at 1 mg/mL. Nitrocellulose membranes were blocked in 10mM sodium phosphate containing bovine serum albumin (BSA), polyvinylpyrrolidone (PVP-40) and sucrose, and dried at 37˚C for 30 minutes. Detection mAb was passively adsorbed to 40 nm colloidal gold particles (DCN Diagnostics, Carlsbad, CA), blocked and concentrated to optical density = 10 at 540 nm. Gold-labeled antibody was sprayed (10 µL/cm) onto Fusion 5 conjugate pad (GE Healthcare, Piscataway, NJ) for the GP-specific LFI prototype or Glass Fiber Standard 14 conjugate pad (GE Healthcare, Piscataway, NJ) for the VP40-specific LFI prototype. LFIs were assembled with a conjugate pad, nitrocellulose membrane and a wicking pad on an adhesive backing card, with each section overlapping to allow for adequate capillary flow. LFIs were cut to 4 mm per strip and stored in foil pouches with desiccants.

### LFI EBOV testing–cultured virus

All BSL4 containment experiments were performed at the Canadian Science Centre for Human and Animal Health, part of the National Microbiology Laboratory (NML), Public Health Agency of Canada (PHAC) in Winnipeg, MB, Canada. EBOV isolate Ebola virus/*H. sapiens*-wt/GIN/2014/Makona-C07 (order *Mononegavirales*, family *Filoviridae*, species *Z. ebolavirus*; GenBank accession no. KJ660347.2) was grown in Vero E6 cells with DMEM supplemented with 2% FBS post-infection [27]. Cell supernatant was collected when cytopathic effects reached 80%. Viral titer of the supernatant was quantified using a TCID50 assay as previously described [28]. After quantification, final concentration of FBS was brought up to 10% and the supernatant was stored in liquid nitrogen. The GP-specific and VP40-specific LFI prototypes were tested side-by-side with the ReEBOV Antigen Rapid Test (Zalgen Labs, Germantown, MD). Two-fold dilutions of supernatant containing live EBOV-Makona from Vero E6 infected cells were thawed and diluted in chase buffer and tested on the LFI, with 40 μL of sample added to the conjugate pad followed by 150 μL of chase buffer. The same volume of sample was tested on the ReEBOV Antigen Rapid Test and the assay was performed otherwise according to the manufacture's guidelines. As a negative control, 40 μL of chase buffer was added to the conjugate pad followed by 150 μL of chase buffer. All assays were visually evaluated after 15–25 minutes.

### Ebolavirus species LFI Testing

Ebolavirus species LFI testing (S3 Fig) was performed at USAMRIID. Testing was done using inactivated tissue culture supernatants (TCS) from indicated viral cultures. After propagation in Vero76 (ATCC) cells, tissue culture supernatants were clarified and tittered by neutral red plaque assay to determine infectious concentration (PFU/mL). Filovirus TCSs were then inactivated by a combination of gamma irradiation and Beta-propiolactone (BPL) treatment. Inactivated filovirus TCS was diluted in PBST (PBS with 0.05% Tween) as indicated and 40 μL of each dilution was applied to the sample pad followed by addition of 150 μL of chase buffer. LFIs were then incubated at room temp for 15 minutes prior to results being visually inspected and images taken. Only devices with a positive control line were counted as valid. PBST was applied as a negative control.

### LFI EBOV testing–NHP samples

Remnant archived serum samples from two rhesus macaques challenged with EBOV were made available by co-authors at the NML, PHAC, Winnipeg, MB, Canada. Briefly, NHPs were infected with Ebola virus/*H. sapiens*-wt/GIN/2014/Makona-C07 by intramuscular injection with 1000 x TCID50 as previously described [27, 28]. Serum samples were taken at multiple intervals after infection and viral titer was quantified using RT-qPCR [28]. The GP-specific LFI prototype, VP40-specific LFI prototype and ReEBOV Antigen Rapid Test were tested side-by-side with the serum samples collected from EBOV infected NHPs. Assays were performed and evaluated as described above. As a negative control, 40 μL of un-infected NHP serum was added to the conjugate pad followed by 150 μL of chase buffer.

### Ethics statement

Laboratory studies performed for this project that utilized mice for mAb production were approved by the University of Nevada, Reno Institutional Animal Care and Use Committee (Protocol # 00024). All work with mice was performed in association with University of Nevada, Reno Office of Laboratory Animal Medicine, which follows the National Institutes of Health Office of Laboratory Animal Welfare policies (Assurance # A3500-01). The infected NHP samples were from a previous animal study that was performed in BSL4 biocontainment

at the Canadian Science Centre for Human and Animal Health, PHAC and approved by the institutional Animal Care Committee following the guidelines of the Canadian Council on Animal Care. Animals were acclimatized for 7 days prior to infection. Animals were fed and monitored twice daily (pre- and post-infection) and fed commercial monkey chow, treats and fruit. Husbandry enrichment consisted of commercial toys and visual stimulation.

## Results

### EBOV mAb production and reactivity

Female CD1 mice were immunized with eVLPs that had been sonicated to allow for sterile filtration prior to immunizations (nano-eVLPs) or purified rVP40. Thirty-two EBOV mAbs were isolated and are described in Table 1. MAb identifiers that begin with "1" were isolated

**Table 1. Summary of GP and VP40 mAbs generated for this study.**

| mAb | Immunization | Subclass* | Reactivity[‡] | Zaire Mayinga Reactivity[‡] | Zaire Guinea reactivity[‡] | Other Ebolavirus species[‡#] |
|---|---|---|---|---|---|---|
| 1HK1 | Nano-eVLP | IgG2b | GP | + | + | + |
| 1HK2 | Nano-eVLP | IgG1 | GP | + | + | - |
| 1HK3 | Nano-eVLP | IgG1 | GP | + | + | - |
| 1HK4 | Nano-eVLP | IgG2b | GP | + | + | - |
| 1HK5 | Nano-eVLP | IgG2b | GP | + | + | - |
| 1HK6 | Nano-eVLP | IgG2a | VP40 | + | - | - |
| 1HK7 | Nano-eVLP | IgG2b | GP | + | + | + |
| 1HK8 | Nano-eVLP | IgG2b | VP40 | + | + | - |
| 1HK9 | Nano-eVLP | IgG1 | GP | + | + | - |
| 1HK10 | Nano-eVLP | IgG1 | GP | + | + | - |
| 1HK11 | Nano-eVLP | IgG1 | GP | + | + | - |
| 1HK12 | Nano-eVLP | IgG2b | VP40 | + | - | - |
| 1HK13 | Nano-eVLP | IgG2a | GP | + | + | - |
| 1HK14 | Nano-eVLP | IgG2b | GP | + | + | - |
| 1HK15 | Nano-eVLP | IgG2b | GP | + | + | - |
| 1HK16 | Nano-eVLP | IgG1 | GP | + | + | - |
| 1HK17 | Nano-eVLP | IgG1 | GP | + | + | - |
| 2HK1 | rVP40 | IgG1 | VP40 | + | + | + |
| 2HK2 | rVP40 | IgG2b | VP40 | + | - | - |
| 2HK3 | rVP40 | IgG2b | VP40 | + | - | - |
| 2HK4 | rVP40 | IgG2b | VP40 | + | - | - |
| 2HK5 | rVP40 | IgG2a | VP40 | + | - | - |
| 2HK6 | rVP40 | IgG2b | VP40 | + | - | - |
| 2HK7 | rVP40 | IgG2b | VP40 | + | - | - |
| 2HK8 | rVP40 | IgG2b | VP40 | + | - | - |
| 2HK9 | rVP40 | IgG2b | VP40 | + | - | - |
| 2HK10 | rVP40 | IgG2b | VP40 | + | - | - |
| 2HK11 | rVP40 | IgG2b | VP40 | + | - | - |
| 2HK12 | rVP40 | IgG2b | VP40 | + | - | - |
| 2HK13 | rVP40 | IgG2b | VP40 | + | - | - |
| 2HK14 | rVP40 | IgG2b | VP40 | + | - | - |
| 2HK15 | rVP40 | IgG2a | VP40 | + | - | - |

* subclass determined by indirect ELISA

[‡] mAb reactivity determined by Western immunoblot

[#] Western blot analysis with ebolavirus species Sudan Boniface, Täi Forest, Bundibuyo, Reston

from mice immunized with nano-eVLPs and mAb identifiers that begin with "2" were isolated from mice immunized with rVP40. An ELISA was performed to determine reactivity with eVLPs and ascertain antibody IgG subclass; all mAbs were determined to contain a $\kappa$ light chain and were either IgG1, IgG2b or IgG2a subclass. MAbs were evaluated by Western immunoblot using recombinant GP (rGP) or rVP40 to determine EBOV antigen reactivity. Fourteen of seventeen mAbs isolated from nano-eVLP immunizations were reactive with GP, however, this immunogen did produce three mAbs reactive with VP40. Western immunoblot results display GP reactivity of mAbs 1HK1, 1HK2, 1HK3, 1HK4, 1HK5, 1HK7, 1HK9, 1HK10, 1HK11, 1HK13, 1HK14, 1HK15, 1HK16 and 1HK17 (Fig 1A) and VP40 reactivity of mAbs 1HK6, 1HK8, 1HK12, 2HK1, 2HK2, 2HK3, 2HK4, 2HK5, 2HK6, 2HK7, 2HK8, 2HK9, 2HK10, 2HK11, 2HK12, 2HK13, 2HK14 and 2HK15 (Fig 1B and Table 1). Reactivity of mAbs with eVLPs was also confirmed by Western immunoblot (S1 Fig). Interestingly, a roughly 80 kD band was most reactive when the VP40 mAbs were used probe the VLPs suggesting that these experimental conditions preserve the dimeric form of VP40 which has been shown to form oligomers as well [29–31]. Additonal Western blots were performed with lysates from other ebolaviruses acquired from BEI Resources (Sudan Boniface—cat # NR-49810, Reston— cat # NR49811, Zaire Mayinga—cat # NR-49809, Zaire Guinea—cat # NR-49462, Täi Forest— cat # NR-49812, Bundibugyo—cat # NR-49813) and the reactivity is summarized in Table 1. The GP specific mAbs were reactive with both Zaire species, however the VP40 mAbs appeared to only be reactive to Zaire Mayinga. There was minimal GP and VP40 mAb reac-tivty non-Zaire Ebola species.

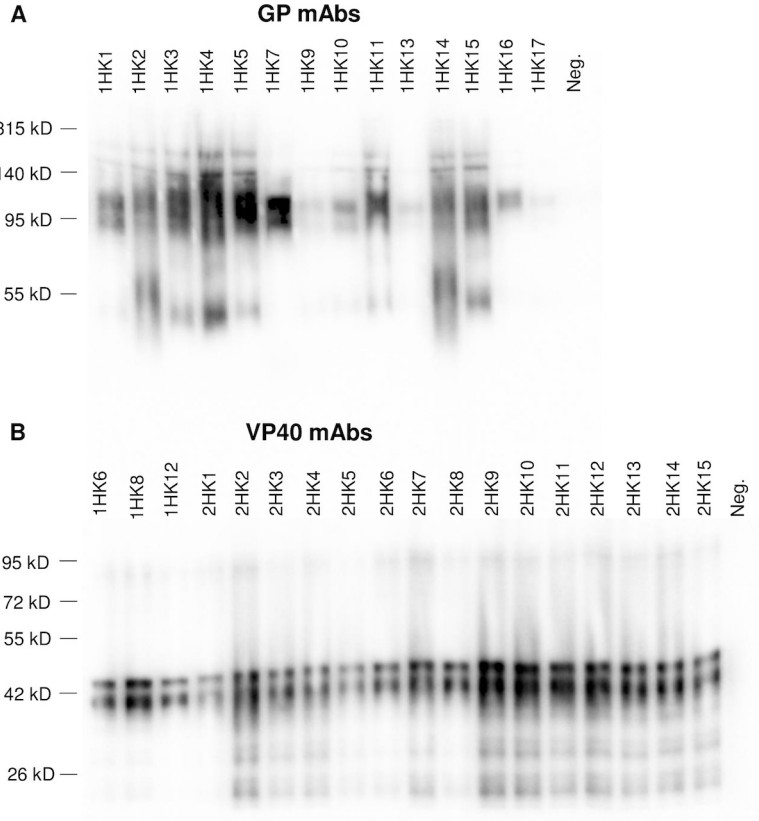

**Fig 1. Western immunoblot analysis to evaluating monoclonal antibody (mAb) reactivity.** mAb reactivity was assessed using 1 μg recombinant GP (Panel A) or 0.5 μg recombinant VP40 (Panel B).

## LFI screening of mAbs

For successful LFI development, it is essential to evaluate mAbs in the LFI platform to determine which mAbs perform optimally in this immunoassay format. Additionally, antigen capture of protein targets requires two antibodies be used as a pair in the immunoassay. Binding to different epitopes on the target protein, one antibody acts to "capture" the target and another labeled antibody allows for "detection". To determine which mAb pairs function best together in the LFI, all combinations were evaluated for capture or detection within each antigen reactive group (GP or VP40). LFIs were initially tested with a standard concentration of eVLPs spiked in buffer or buffer alone as a negative control. Selection of mAb pairs for further screening was based on highest reactivity with eVLPs and minimal to no background signal with buffer alone. Next, LFIs were screened for eVLP reactivity and background by spiking eVLPs into normal human serum (NHS) and testing NHS alone as a negative control. MAb pairs were eliminated if high background was visually observed when testing with NHS. To aid the visual assessment of assays, a Qiagen ESE lateral flow reader was used to obtain a readout for test line reactivity and background signal. Top LFI mAb pairs were selected from each antigen reactive group based on the highest test line signal minus test line background when tested in buffer. One result for each prototype is listed in Table 2 as this was done to rapidly downselect top performing pairs. For GP mAbs the top ten LFI mAb pairs contained 1HK1, 1HK3, 1HK4, 1HK5, 1HK7, 1HK11, 1HK15 and 1HK16. For VP40 mAbs the top ten LFI mAb pairs contained 1HK6, 1HK8, 1HK12, 2HK1, 2HK2, 2HK7 and 2HK12.

## Antigen-capture ELISA

These highest ranked mAbs were then analyzed by antigen-capture ELISA to further assess mAb combinations and rank the performance of selected GP and VP40 mAb pairs for

**Table 2. Reactivity of top performing monoclonal antibody (mAb) pairs when evaluated in lateral flow immunoassay (LFI) format with target antigen spiked into buffer.**

| Antigen | Rank | Capture mAb | Detection mAb | Test Line Background (mm*mV) | Test Line Signal (mm*mV) | Signal minus Background (mm*mV) |
|---|---|---|---|---|---|---|
| GP LFIs | 1 | 1HK7 | 1HK3 | 0 | 608 | 608 |
| | 2 | 1HK7 | 1HK11 | 31 | 555 | 524 |
| | 3 | 1HK11 | 1HK7 | 0 | 339 | 339 |
| | 4 | 1HK7 | 1HK5 | 0 | 315 | 315 |
| | 5 | 1HK3 | 1HK7 | 0 | 304 | 304 |
| | 6 | 1HK7 | 1HK4 | 0 | 300 | 300 |
| | 7 | 1HK1 | 1HK11 | 159 | 425 | 266 |
| | 8 | 1HK4 | 1HK7 | 59 | 318 | 259 |
| | 9 | 1HK3 | 1HK16 | 144 | 387 | 243 |
| | 10 | 1HK7 | 1HK15 | 0 | 207 | 207 |
| VP40 LFIs | 1 | 2HK12 | 2HK1 | 53 | 675 | 622 |
| | 2 | 2HK2 | 2HK1 | 79 | 628 | 549 |
| | 3 | 2HK1 | 2HK7 | 0 | 541 | 541 |
| | 4 | 2HK1 | 2HK12 | 0 | 505 | 505 |
| | 5 | 2HK1 | 1HK12 | 0 | 487 | 487 |
| | 6 | 2HK7 | 2HK1 | 63 | 538 | 475 |
| | 7 | 2HK1 | 2HK1 | 0 | 456 | 456 |
| | 8 | 2HK1 | 1HK8 | 0 | 445 | 445 |
| | 9 | 1HK8 | 2HK1 | 82 | 524 | 442 |
| | 10 | 1HK6 | 2HK1 | 288 | 716 | 428 |

**Table 3. Limit of detection (LOD) of optimized antigen-capture ELISA with Ebola virus-like particles (eVLPs) and purified GP and VP40 recombinant antigen (ng/mL ± SD).**

|  | Capture | Detection | eVLP LOD | Antigen LOD |
|---|---|---|---|---|
| **GP mAbs** | 1HK7 | 1HK11 | 10 ± 2.1 | 0.37 ± 0.06 |
|  | 1HK7 | 1HK4 | 6.0 ± 0.64 | 0.23 ± 0.09 |
| **VP40 mAbs** | 2HK1 | 2HK7 | 11 ± 1.6 | 12 ± 1.5 |
|  | 2HK12 | 2HK1 | 7.7 ± 1.1 | 1.4 ± 0.10 |

detection of their viral target (S1 Table). It is important to note that the initial antigen-capture ELISAs were done using standard antibody coating and detection concentrations merely for ranking of mAb pairs. The data in S1 Table should not be construed as a true limit of detection (LOD), but should be interpreted as preliminary LODs for the purposes of selecting mAb pairs for advancement. GP mAb pairs showed a wide range of sensitivity, with mAb pair 1HK4 (capture):1HK11 (detection) showing the best sensitivity. However, this mAb pair was not one of the better performing pairs in the LFI format, so it was ruled out for further testing. Two other GP mAb pairs that performed well in the antigen capture ELISA and also in LFI format were 1HK7 (capture):1HK4 (detection) and 1HK7 (capture):1HK11 (detection). A tighter range of sensitivity was observed with the VP40 mAbs, with 2HK1 (capture):2HK7 (detection) and 2HK12 (capture):2HK1 (detection) performing the best. Both of these mAb pairs also performed well in the LFI format with high reactivity and no detectable background (Table 2).

Antigen-capture ELISA conditions were further optimized with the two best GP or VP40 mAb pairs to determine more accurate LOD levels. Optimized assays were then used to determine LODs with eVLPs and recombinant antigens (Table 3). With optimized conditions, the GP mAb pairs 1HK7 (capture):1HK11 (detection) and 1HK7 (capture):1HK4 (detection) showed improved detection of eVLPs and high sensitivity for rGP; LODs of 0.37 ng/mL and 0.23 ng/mL, respectively. The top two VP40 mAb pairs also showed improved sensitivity with eVLPs after optimization of assay conditions. The LODs with rVP40 were 12 ng/mL with 2HK1 (capture):2HK7 (detection) and 1.4 ng/mL with 2HK12 (capture):2HK1 (detection). Results were noted that the latter pair may be a preferred candidate for LFI prototype development given the > 8-fold lower LOD.

## Binding affinity of mAbs by surface plasmon resonance (SPR)

MAbs that performed well in the standard LFI and optimized antigen-capture ELISA formats (GP: 1HK4, 1HK7 and 1HK11 and VP40: 2HK1, 2HK7 and 2HK12) were selected for kinetic analysis using SPR with nano-eVLPs (S2 Fig). SPR experiments were performed in triplicate for each antibody and the data was analyzed with BIAevaluation software using a bivalent analysis. The average association rate ($k_a$), dissociation rate ($k_d$), and affinity ($K_D$; $K_D = k_d/k_a$) of each mAb was determined and is reported in Table 4. GP mAbs all displayed similar fast association rates between $3.4 \times 10^5$ and $5.9 \times 10^5$ $M^{-1}s^{-1}$. However, the mAbs were distinguished by a difference in dissociation rates, with 1HK7 dissociating nearly 100-fold slower than 1HK4 and 1HK11. As a result, 1HK7 has the highest affinity (lowest $K_D$) out of the three mAbs with a $K_D$ = 0.4 nM. The VP40 mAbs (2HK1, 2HK7 and 2HK12) have comparable association rates ($> 10^5$ $M^{-1}s^{-1}$) and dissociation rates ($10^{-4}$–$10^{-5}$ $s^{-1}$), resulting in high affinity values, $K_D < 1$ nM, for all three mAbs. Antibody performance can vary between immunoassays, but the high affinity of these mAbs determined by SPR analysis supports their potential as key reagents for developing a sensitive, rapid EBOV diagnostic.

**Table 4. Surface plasmon resonance analysis of EBOV monoclonal antibody (mAb) binding to nano-Ebola virus-like particles.**

|            | mAb   | $k_a$ ($M^{-1}s^{-1}$) | $k_d$ ($s^{-1}$) | $K_D$ (nM) |
|------------|-------|-------------------------|-------------------|-------------|
| **GP mAbs** | 1HK4  | 4.1E+05                 | 2.0E-02           | 48          |
|            | 1HK7  | 5.9E+05                 | 2.6E-04           | 0.4         |
|            | 1HK11 | 3.4E+05                 | 1.3E-02           | 39          |
| **VP40 mAbs** | 2HK1  | 1.8E+05              | 1.6E-04           | 0.9         |
|            | 2HK7  | 2.1E+05                 | 5.3E-05           | 0.3         |
|            | 2HK12 | 1.6E+05                 | 6.0E-05           | 0.4         |

## LFI prototype development and reactivity with live EBOV

Following the mAb evaluation studies that included LFI screening, antigen-capture ELISA and SPR analysis, prototype LFIs were developed and optimized utilizing the top two mAb pairs to each target antigen. Final screening of optimized LFIs, that included further evaluation of visual LOD, background and matrix effects when testing NHS, resulted in selection of 1HK7 (capture):1HK11 (detection) for the GP prototype and 2HK12 (capture):2HK1 (detection) for the VP40 prototype.

The final prototypes were then tested with cultured EBOV to verify reactivity with native antigens and to assess sensitivity. GP-specific and VP40-specific LFI prototypes were evaluated in a biosafety level 4 (BSL4) laboratory for reactivity with live EBOV (variant Makona) that was grown in Vero E6 cells. Additionally, a commercially available rapid EBOV diagnostic assay, ReEBOV Antigen Rapid Test (Zalgen Labs, Germantown, MD), was tested in parallel for comparison. The ReEBOV Antigen Rapid Test received Emergency Use Authorization from the FDA during the 2013–2016 outbreak and is designed to detect EBOV VP40 [20, 32]. Dilutions of EBOV infected Vero cell supernatant were tested side-by-side on the three LFIs. Results show the EBOV GP-specific LFI prototype to have the highest sensitivity, with a visual LOD = 1.2 x $10^4$ TCID50/mL (Fig 2A). The EBOV VP40-specific LFI prototype is less sensitive than the EBOV GP-specific LFI prototype, with a LOD = 9.8 x $10^4$ TCID50/mL (Fig 2B), yet still slightly more sensitive than the comparator assay that also detects VP40 (LOD = 2.00 x $10^5$ TCID50/mL) (Fig 2C). Overall, these results showed the EBOV GP-specific LFI prototype to have a > 16-fold improvement in sensitivity compared to the ReEBOV Antigen Rapid Test when testing in vitro cultured EBOV.

## LFI prototype reactivity with EBOV infected NHP serum samples

Availability of remnant serum samples from a prior study of EBOV infected NHPs allowed for an assessment of clinical sensitivity. Samples collected at 4, 5, 6, and 7.5 days post-infection (dpi) from two NHPs infected with EBOV (variant Makona) were used to evaluate the sensitivity of the EBOV GP-specific LFI prototype, EBOV VP40-specific LFI prototype and ReEBOV Antigen Rapid Test (Fig 3A). The EBOV GP-specific LFI prototype displayed a visible test line at 4 dpi, while the EBOV VP40-specific LFI prototype and ReEBOV Antigen Rapid Test were negative at this early time point. All assays were positive by 5 dpi, although there was noticeable variability between test line intensity. Individuals reading the assays also reported some streaking on the membranes of the two prototype assays which is evident in the images presented in Fig 3. This streaking was not observed with positive and negative buffer controls and was attributed to effects of serum as the matrix.

As a part of an independent broader study, RT-qPCR was performed on the samples collected at 3, 4, 5, 6, 7, and 7.5 dpi from the same two NHPs. Although samples collected 3 and 7

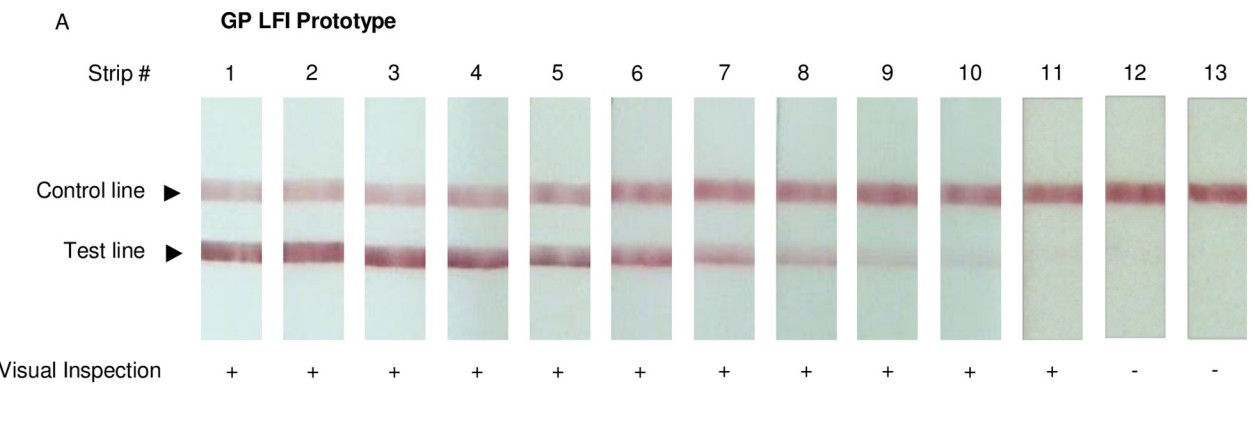

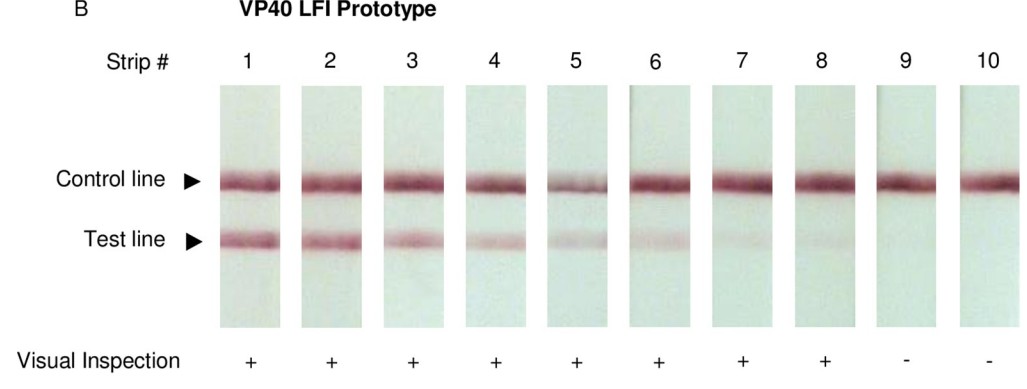

| Strip # | TCID50/mL |
|---------|-----------|
| 1 | 1.30E+07 |
| 2 | 6.30E+06 |
| 3 | 3.20E+06 |
| 4 | 1.60E+06 |
| 5 | 7.90E+05 |
| 6 | 3.90E+05 |
| 7 | 2.00E+05 |
| 8 | 9.80E+04 |
| 9 | 4.90E+04 |
| 10 | 2.50E+04 |
| 11 | 1.20E+04 |
| 12 | 6.20E+03 |
| 13 | 3.10E+03 |

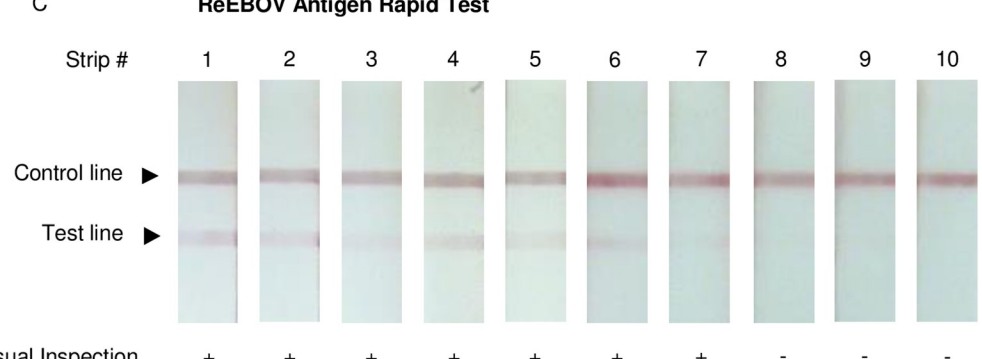

**Fig 2. Sensitivity of EBOV lateral flow immunoassay (LFI) with live cultured EBOV.** The GP-specific LFI prototype, VP40-specific LFI prototype and ReEBOV Antigen Rapid Test (Zalgen Labs) were evaluated for sensitivity with two-fold dilutions of EBOV (variant Makona) culture supernatant; positive (+) or negative (-) results are reported below each LFI. Viral titer of the supernatant was quantified using a TCID50 assay.

dpi were not available for testing of LFIs, the results show that the EBOV GP-specific LFI detected infection in both animals at the same time PCR became positive at 4 dpi (Fig 3B). Quantitative viremia results for the 4 dpi samples were 1.6 x $10^3$ GEQ/mL (NHP #1) and 2.4 x $10^4$ GEQ/mL (NHP #2).

The GP-specific prototype was performing well when tested against a Zaire Makona isolate. A small batch of LFIs was sent to USAMRIID for testing against a variety of Ebola virus species. As shown in S3 Fig, the GP-specific LFI appears to be specific to Zaire species only.

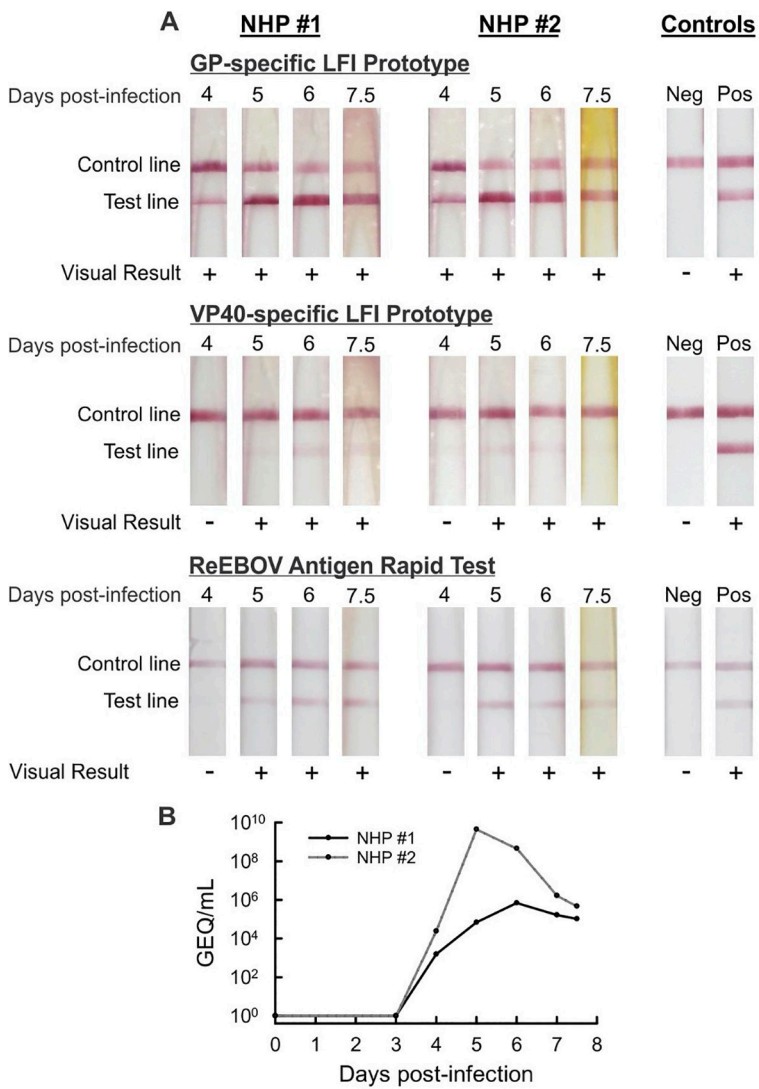

**Fig 3. EBOV lateral flow immunoassay (LFI) prototype testing of infected non-human primate (NHP) serum.**
Panel A–The GP-specific LFI prototype, VP40-specific LFI prototype and ReEBOV Antigen Rapid Test (Zalgen Labs)
were tested with serum samples taken 4, 5, 6, and 7.5 days post-infection (DPI) from two EBOV (variant Makona)
infected NHPs; positive (+) or negative (-) results are reported below each LFI. Panel B–Quantitative viremia
determined via qRT-PCR for 0, 3, 4, 5, 6, 7, and 7.5 days dpi from the same two NHPs; results are reported as genome
equivalents (GEQ) per mL of sample.

Reactivity was clear for two isolates of Zaire Mayinga, however, reactivity was very weak or
undetectable against two Sudan isolates and one Reston, and Bundibugyo species. The LFIs
shown in S3 did not run optimally as there is visable streaking of gold-conjugate, however, the
two Zaire Mayinga isolates that were tested produced very strong reactivity at a 1 x10$^6$ PFU/ml
dilution and appeared to also be positive at the 1–5 x 10$^4$ PFU/ml dilutions.

## Discussion

The need for a more sensitive RDT for diagnosis of EBOV was never more evident than during
the 2013–2016 Ebola outbreak and was the impetus for the current study. The production of a
large panel of mAbs followed by thorough screening and evaluation of performance in

multiple immunoassay platforms enabled the characterization and selection of highly reactive mAb pairs for development of rapid test prototypes to two different EBOV targets.

All mAbs showed reactivity to EBOV antigens by initial screening via different immunoassay platforms. While these results provided helpful guidance for selection of initial hybridoma clones and yielded additional insight into antibody-antigen interactions, it was essential to evaluate antibodies in the immunoassay platform of interest. Optimal LFI design parameters for individual mAb pairs often varies greatly. However, the impracticality of optimizing conditions for all mAb pairs dictated that initial screening be conducted using a standardized LFI design for ranking and down selection; more than 2,100 LFIs were screened. Given that multiple mAb pairs showed promise and that final selection should not be based solely on results from un-optimized assays, additional mAb characterization was performed to narrow down the number of mAb pairs to be considered for prototype development. Initial antigen-capture ELISA results provided added data on the reactivity of mAb pairs and aided in further selection. More advanced optimized antigen-capture ELISAs using chosen mAb pairs allowed for determining LODs with both eVLPs and recombinant antigens. The ELISA LOD analysis yielded encouraging results with detection of GP in the pg/mL range and VP40 in low ng/mL. These results supported selection of these mAbs for SPR analysis to determine kinetic properties of the antibody interactions with their target antigens.

Antibody affinity for binding to protein targets is an important factor that contributes to assay performance. Equally important is the ability for two antibodies to work in tandem, binding to two separate epitopes in a conformational manner conducive to efficient capture and detection of the target. As a result, the best pair may not necessarily be the two antibodies with highest affinity, but will be one where mAb affinity and the binding specificity of each antibody function in harmony to allow for the most sensitivity detection of the diagnostic target. The SPR kinetic results from the three GP mAbs provides an example of this, where one mAb (1HK7) has a 100-fold greater affinity compared to its pair (1HK11), but the two antibodies paired together demonstrated superior sensitivity.

After final selection of mAb pairs for each target and production of optimized assays, studies advanced to BSL4 testing of live cultured EBOV. This testing was considered essential for confirming reactivity of the LFI prototypes with native EBOV antigen and evaluating sensitivity. Results demonstrated the EBOV GP-specific LFI prototype had a > 16-fold improvement in sensitivity over the ReEBOV Antigen Rapid Test that detects VP40. The EBOV VP40-specific LFI prototype developed as a part of this study showed a nominal 2-fold improvement over the comparator assay. Given the data was from in vitro studies, the results were interpreted with cautious optimism.

As mentioned previously, another key factor for a successful diagnostic test is the expression and availability of disease targets in clinical samples. The rapid onset and progression of EVD symptoms dictates the need for early diagnosis when concentrations of EBOV targets are low. It is known that both GP and VP40 are present during EVD and are considered relevant diagnostic targets, yet it was unclear at the beginning of this study which target may be superior for immunoassay detection. We did isolate many high affinity VP40 mAbs and VP40 is abundantly expressed. However, this did not translate into production an LFI that was more sensitive when testing was performed on virus isolated from tissue culture compared to the GP prototype. One possibility could be with the reduced availability of VP40 as a target when produced as a component of VLPs or virions in tissue culture or *in vivo*. As mentioned, VP40 can form dimers and oligomers [29–31] which may prevent efficient binding of reactive mAbs. Also, it could be possible that our GP prototype is detecting the soluble truncated form of GP as well. This soluble GP (sGP) is 90% homologous with the N-terminus of full-length GP and has been shown to be highly expressed [33–36]. sGP is abundantly secreted from infected cells,

and therefore may be an important diagnostic target [37–39]. While the results of the EBOV GP-specific LFI prototype with cultured EBOV were considered encouraging, it would be naïve to base further decisions solely on outcomes from in vitro testing. It must always be considered that results from in vitro growth conditions may not translate to results of clinical testing, as in vivo expression levels in clinical samples have the potential to be quite different.

Complexities associated with obtaining clinical samples for EBOV research and development is a limiting factor for many researchers. The NHP serum samples in this study were restricted to four time-course samples from only two animals. However, the opportunity to test samples from a NHP model of EBOV infection provided crucial information for evaluation of the LFI prototypes. The results of the EBOV GP-specific LFI prototype detecting infection 24 hours earlier than the two VP40 assays (4 dpi vs 5 dpi in both animals), clearly demonstrates the greater sensitivity of the GP-specific LFI prototype. RT-qPCR results provide additional insight into the sensitivity and potential utility of the EBOV GP-specific LFI prototype, showing that the prototype RDT was able to detect infection at the same time as the highly sensitive molecular assay, with a positive LFI result at the lowest level of EBOV detected by PCR; NHP #1 at 4 dpi. Given the relatively strong signal observed with this sample, it is hypothesized that the GP-specific LFI may be capable of detecting even lower levels of EBOV. However, this information is unknown, as 4 dpi was the earliest time point available for LFI testing. Additionally, the RT-qPCR results align with and support the difference observed in the intensity of the GP-specific LFI prototype test lines, with the higher level of viremia in the 4 dpi NHP #2 sample resulting in a stronger test line signal than that observed with the NHP #1 sample from the same time point.

Visual assessment of the prototype assays also showed the need for further optimization to resolve streaking observed when testing some serum matrices. Initial optimization did include evaluation with pooled human serum from healthy individuals, but a valuable lesson was learned that future LFI design testing needs to include a greater number of lots of both normal and diseased sera from multiple commercial sources.

Currently, there are two Ebola RDTs approved for use by the FDA. Emergency use authorization (EUA) of ReEBOV rapid antigen detect assay was recently revoked due to FDA cited issues with performance [40]. The OraQuick Ebola rapid antigen has reached De Novo, 510(k) approval, it is a pan-reactive ebolavirus assay that detects VP40. An LoD value of $1.64 \times 10^6$ TCID50/mL inactivated Zaire Mayinga stock spiked into whole blood was reported in the product insert for the Oraquick RTD. Chembio DPP Ebola antigen system has an emergency use authorization designation from the FDA, it is Zaire reactive and detects VP40. An LOD value of $1.23 \times 10^5$ TCID50/ml Zaire Makona spiked into whole blood was reported in their product insert. Our GP prototype LFI does not appear to be pan-reactive to Ebola virus species, however it is quite analytically sensitive with Ebola Zaire species, the most common cause of outbreaks. In the early stages our development our RDT appears able to detect virus at $1.2 \times 10^4$ TCID50/ml spiked into serum. In patients, upon the initial days of EVD symptoms, the viral load can range from $5 \times 10^1$ to $7 \times 10^7$ pfu/mL in serum from EBOV infected patients [19–21]. When testing our NHP samples we have shown there is potential to identify positive samples at $1.6 \times 10^3$ GEQ/mL; therefore, we hope that further development of this GP prototype will result in the earlier identification and isolation of patients with lower levels of viremia.

Our study described the development of a large panel of EBOV GP and VP40 reactive mAbs that led to the development of encouraging protype ELISAs and LFIs. We do feel as though development of the LFI is critical as there is a great need for POC testing for EVD. However, if clinical microbiology laboratories are available in endemic regions, a sensitive antigen-capture ELISA may be very useful in a setting where hundreds of samples need to be analyzed at once.

In conclusion, testing of samples from a NHP model of EBOV infection showed one GP-specific LFI prototype developed in this study was capable of detecting EBOV i) as early as RT-qPCR, ii) at least 24 hours earlier than the comparator RDT and iii) with a analytical sensitivity of at least $1.6 \times 10^3$ GEQ/ml. Additional required studies include specificity testing of a larger panel of Ebolavirus isolates, even though the current GP prototype appears to reactive with EBOV Zaire species. Future studies will also include testing of a panel of microbes that cause similar symptoms to Ebola in the same geographic region (e.g. Lassa fever virus, leptopira, plasmodium, etc.). In addition, it is neccesary to test additional samples from NHPs and EBOV infected patients in order to firmly establish that the current GP prototype should evaluated for FDA approval. Our studies to date greatly support the need for further development, as LFI design, buffer components and even the mAbs may change. To our benefit we have an encouraging panel of mAbs to choose from if needed. Development efforts are ongoing and driven by the encouraging results that demonstrate the potential of a RDT capable of earlier detection of EVD.

## Supporting information

**S1 Fig. Western immunoblot analysis evaluating EBOV monoclonal antibody (mAb) reactivity.** GP mAbs (Panel A) and VP40 mAbs (Panel B) were assessed for reactivity using Ebola virus-like particles (1 ug).
(TIFF)

**S2 Fig. Surface plasmon resonance analysis of monoclonal antibody (mAb) binding to nano-eVLPs.** Panel A–GP mAbs: 1HK4, 1HK7 and 1HK11. Panel B–VP40 mAbs: 2HK1, 2HK7 and 2HK12. Data shown is from a single experiment representative of three independent experiments.
(TIFF)

**S3 Fig. Ebola virus species tested for reactivity on GP prototype.** Dilutions of inactivated Ebola virus species (e.g. 1E6 = $1 \times 10^6$ PFU/ml) were tested on a GP LFI prototype. This prototype was developed using 1HK7 (capture):1HK11 (detection) GP mAbs. The sample did not run optimally, note the streaking of gold labeled 1HK11 below the control (highest reactive line on each LFI). Empty areas within the figure indicate that these dilutions were not analyzed.
(TIFF)

**S1 Table. Preliminary EBOV antigen-capture ELISAs evaluating sensitivity of GP and VP40 mAb pairs with virus-like particles (ng/ml).**
(DOCX)

## Acknowledgments

We are grateful to CL4 animal care staff Kevin Tierney and veterinary staff of NML, PHAC, for their support with the NHP study.

## Author Contributions

**Conceptualization:** Denise F. Reyes, Marcellene A. Gates-Hollingsworth, Xiangguo Qiu, David P. AuCoin.

**Data curation:** Haley L. DeMers, Shihua He, Sujata G. Pandit, Emily E. Hannah, Zirui Zhang, Feihu Yan, Heather R. Green, Denise F. Reyes, Derrick Hau, Megan E. McLarty.

**Formal analysis:** Haley L. DeMers, Shihua He, Sujata G. Pandit, Zirui Zhang, Feihu Yan, Heather R. Green, Denise F. Reyes, Derrick Hau, Megan E. McLarty, Marcellene A. Gates-Hollingsworth.

**Funding acquisition:** Xiangguo Qiu, David P. AuCoin.

**Investigation:** Haley L. DeMers, Shihua He, Sujata G. Pandit, Emily E. Hannah, Zirui Zhang, Feihu Yan, Heather R. Green, Denise F. Reyes, Derrick Hau, Megan E. McLarty, Louis Altamura, Cheryl Taylor-Howell, Marcellene A. Gates-Hollingsworth, Xiangguo Qiu, David P. AuCoin.

**Methodology:** Haley L. DeMers, Shihua He, Sujata G. Pandit, Emily E. Hannah, Zirui Zhang, Feihu Yan, Heather R. Green, Denise F. Reyes, Derrick Hau, Megan E. McLarty, Marcellene A. Gates-Hollingsworth, Xiangguo Qiu, David P. AuCoin.

**Project administration:** Shihua He, Marcellene A. Gates-Hollingsworth, Xiangguo Qiu, David P. AuCoin.

**Resources:** Xiangguo Qiu, David P. AuCoin.

**Software:** Xiangguo Qiu, David P. AuCoin.

**Supervision:** Marcellene A. Gates-Hollingsworth, Xiangguo Qiu, David P. AuCoin.

**Validation:** Haley L. DeMers, Shihua He, Sujata G. Pandit, Emily E. Hannah, Zirui Zhang, Feihu Yan, Heather R. Green, Denise F. Reyes, Derrick Hau, Megan E. McLarty, Louis Altamura, Cheryl Taylor-Howell, Xiangguo Qiu, David P. AuCoin.

**Writing – original draft:** Haley L. DeMers, Marcellene A. Gates-Hollingsworth, David P. AuCoin.

**Writing – review & editing:** Haley L. DeMers, Shihua He, Sujata G. Pandit, Emily E. Hannah, Zirui Zhang, Feihu Yan, Heather R. Green, Denise F. Reyes, Derrick Hau, Megan E. McLarty, Marcellene A. Gates-Hollingsworth, Xiangguo Qiu, David P. AuCoin.

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
