## [Decision Letter · Decision Letter 0]

28 Apr 2020

Dear Dr. AuCoin,

Thank you very much for submitting your manuscript "Development of an antigen detection assay for early point-of-care diagnosis of Zaire ebolavirus" for consideration at PLOS Neglected Tropical Diseases. As with all papers reviewed by the journal, your manuscript was reviewed by members of the editorial board and by several independent reviewers. In light of the reviews (below this email), we would like to invite the resubmission of a significantly-revised version that takes into account the reviewers' comments. 

I have read your manuscript very carefully and have decided to make decision "Major Revision". All the reviewers have recommended me to make the decision. I agree with the comments raised by these reviewers. 

We cannot make any decision about publication until we have seen the revised manuscript and your response to the reviewers' comments. Your revised manuscript is also likely to be sent to reviewers for further evaluation.

Sincerely,

Masayuki Saijo

Guest Editor

Samuel Scarpino

Deputy Editor

Reviewer's Responses to Questions

**Key Review Criteria Required for Acceptance?**

**Methods**

-Are the objectives of the study clearly articulated with a clear testable hypothesis stated?

-Is the study design appropriate to address the stated objectives?

-Is the population clearly described and appropriate for the hypothesis being tested?

-Is the sample size sufficient to ensure adequate power to address the hypothesis being tested?

-Were correct statistical analysis used to support conclusions?

-Are there concerns about ethical or regulatory requirements being met?

Reviewer #1: Please see comments below.

Reviewer #2: There is a big difference in the band size of VP40 between Fig. 1 and S1 Fig. This should be explained. 

Lines 132-134: Variant name of the cloned genes (GP and VP40) should be provided.

Reviewer #3: 1. It is not described how to pre-treat EBOV samples before loading onto LFI strip. The information will be important to understand the accessibility of monoclonal antibodies to epitopes of GP or VL40.

**Results**

-Does the analysis presented match the analysis plan?

-Are the results clearly and completely presented?

-Are the figures (Tables, Images) of sufficient quality for clarity?

Reviewer #1: Please see comments below.

Reviewer #2: The major issue with this study is the lack of confirmation for the specificity of the LFI test. Since the Makona variant is phylogenetically distinct from the other Ebola viruses in Zaire ebolavirus species, other variants (e.g., Mayinga, Kikwit, etc.) should also be tested for the sensitivity of the LFI assay. In addition, information on the reactivity of the antibodies to other ebolaviruses (e.g., Sudan and Bundibugyo viruses) should be provided

Considering the practical use, some other viral, bacterial, and protozoal pathogens that cause similar symptoms (other hemorrhagic fever viruses, Leptospira, Plasmodium, etc.) should also be tested to confirm the specificity of the LFI assay.

Reviewer #3: 1. This study did not show epitopes of corresponding monoclonal antibodies. Therefore, the cross-reactivity to other ebolavirus species is unpredictable.

2. The stability of monoclonal antibodies dried in LFI strip was not described. Although the stability is generally recognized, the stability in terms of the reactivity of monoclonal antibodies might be varied among different monoclonal antibodies. 

3. The manuscript mainly described for the development of LFI, while authors described about the antigen-capture ELISA system as well. Although the use of two monoclonal antibodies is overlapped, it is not clear why the demonstration of the antigen-capture ELISA system is required in this manuscript. Unless the sensitivity is compared between antigen-capture ELISA system and LFI, the antigen-capture ELISA description can be moved to supplemental data.

**Conclusions**

-Are the conclusions supported by the data presented?

-Are the limitations of analysis clearly described?

-Do the authors discuss how these data can be helpful to advance our understanding of the topic under study?

-Is public health relevance addressed?

Reviewer #1: Please see comments below.

Reviewer #2: There are some other LFI tests that have been clinically used during the past outbreaks (i.e., OraQuick Ebola, SD Q Line Ebola Zaire Ag, QuikNavi-Ebola, etc.). These should be stated somewhere in the text. The authors should also explain what is the advantage of their LFI assay, compared to the previously developed immunochromatography tests.

Reviewer #3: Due to previously reported LFI, which can broadly detect ebolaviruses, authors in this manuscript should demonstrate which points of LFI in this study could be superior to previous one: cross-reactivity and stability could be important aspects to compare.

**Editorial and Data Presentation Modifications?**

Reviewer #1: Please see comments below.

Reviewer #2: Lines 25, 67, 69, 75, and 83: This outbreak is not called “pandemic”.

Line 423, 2HK1 (capture):2HK12 (detection): Should this be 2HK12 (capture):2HK1 (detection)?

Reviewer #3: 1. Line 344: “if “high background” was observed…. Please clarify the threshold of “high background”.

2. Table 3: Please provide footnote explaining the value of LOD: e.g., mean or SD or SE.

3. Line 423: Data indicated 2HK1 (detection) and 2HK12 (capture) was optimal. Please explain why the opposite combination was selected.

**Summary and General Comments**

Reviewer #1: This paper reported a point-of-care diagnostic test for EBOV with lateral flow immunoassay (LFI) using anti-GP and VP40 monoclonal antibodies newly developed in this study. The LFI is more sensitive than commercially available LFI test kit. The LFI is highly required in the scene of ebolavirus disease outbreak. Therefore improved LFI should be shared by the medical and research community through this journal. However, I concerned several points in this manuscript for publication. I hope my comments will improve the quality of their manuscript.

Major comments:

1. In line 108-109, The authors mentioned that VP40 is most abundantly expressed viral protein. And the results of SPR data in Table 4 showed three anti-VP40 mAbs had higher affinities to the target protein than anti-GP mAbs. However, their assay could detect GP in blood samples more sensitively. I think the authors should add more discussion to explain the reason why LFI using anti-GP mAbs worked better than that using anti-VP40 mAbs. Are there any previous data which show similar results? or did sGP in the infected samples detected in this assay and increased the sensitivity of this assay? 

2. Fig1. Anti-GP mAbs detected multiple or smear bands in western blotting in Figure 1A. The authors should indicate which GP proteins or its derivative (e.g. GP1,2, GP1 or GP2) were detected aside of the figure.

3. Fig3. The author suggested that the samples of NHP#2 in day 7.5 resulted in positive in the result of VP40-specific LFI Prototype. However, I could not find a difference of the band intensity between sample of day 7.5 and that of day 4, which they concluded as negative. 

4. Table 2 SD values for Background and Signal intensity determined with indicated mAbs are not shown in this Table. How many replicated experiments did you try for this result? The authors should add the information, and if necessary, they also add SD for each value.

5. Line 548, The authors hypothesized that the GP-specific LFI many be capable of detecting even lower levels of EBOV, however, there are a lot of reports which determined the viral load in the serum or blood of EVD cases. I suppose they can clearly discuss how effectively the LFI can determine the EVD cases with the clinical samples, and limitation of this assay.

6. S1 Fig. Although the authors used same mAbs for GP and VP40 in Figure 1 and Figure S1, the band size of Figure S1 is not consistent to that of Figure 1. Which size of the band are correct? VP40 is detected around 40kDa, however, the major band appeared at around 80kDa in Figure S1. It looks strange for me.

7. Did the authors examined cross-reactivities of the mAbs for GP or VP40, at least the highly reactive mAbs used in this LFI with other ebolavirus species, e.g. Sudan, Bundibugyo EBOV or other microbes which cause acute febrile illness in the sub-Sahara Africa which is endemic region of EVD, e.g. Plasmodium, Dengue, and Typhi? 

Minor comments:

1. Line 67,69, 83; In general, the word of “pandemic” is not commonly used to represent the outbreak of Ebola in West Africa. I recommend replacing into “Epidemic” as reference 9.

2. Line 405, 100X  100-fold

3. Line 423, 2HK12(capture):2HK1(detection)  2HK12(capture):2HK1(detection)

Reviewer #2: In this study, the authors produced monoclonal antibodies to Ebola virus glycoprotein (GP) and VP40, characterized their properties, and used them to develop a lateral flow immunoassay (LFI) for Ebola virus disease diagnosis. Overall the manuscript is well written. However, the principal conclusion in this study is identical to those of the previous works. The only novel aspect may be the use of anti-GP antibodies for LFI.

Major comments

1. The major issue with this study is the lack of confirmation for the specificity of the LFI test. Since the Makona variant is phylogenetically distinct from the other Ebola viruses in Zaire ebolavirus species, other variants (e.g., Mayinga, Kikwit, etc.) should also be tested for the sensitivity of the LFI assay. In addition, information on the reactivity of the antibodies to other ebolaviruses (e.g., Sudan and Bundibugyo viruses) should be provided.

2. Considering the practical use, some other viral, bacterial, and protozoal pathogens that cause similar symptoms (other hemorrhagic fever viruses, Leptospira, Plasmodium, etc.) should also be tested to confirm the specificity of the LFI assay.

Minor comments

1. There is a big difference in the band size of VP40 between Fig. 1 and S1 Fig. This should be explained. 

2. There are some other LFI tests that have been clinically used during the past outbreaks (i.e., OraQuick Ebola, SD Q Line Ebola Zaire Ag, QuikNavi-Ebola, etc.). These should be stated somewhere in the text. The authors should also explain what is the advantage of their LFI assay, compared to the previously developed immunochromatography tests. 

3. Lines 25, 67, 69, 75, and 83: This outbreak is not called “pandemic”.

4. Lines 132-134: Variant name of the cloned genes (GP and VP40) should be provided.

5. Line 423, 2HK1 (capture):2HK12 (detection): Should this be 2HK12 (capture):2HK1 (detection)?

Reviewer #3: The manuscript entitled “Development of an antigen detection assay for early point-of-care diagnosis of Zaire ebolavirus” by DeMers HL et al. described that the lateral flow immunoassay (LFI) made with a combination of two original monoclonal antibodies against Zaire ebolavirus (EBOV) can detect EBOV antigens in culture supernatants or serum samples of infected nonhuman primates. The point-of-care tool for the detection of EBOV was previously developed in response to the need during EVD outbreak in West Africa. The LFI made by USAMRIID team could detect GP of Zaire, Sudan, Tai Forest, and Reston virus (range: 5x10^4 to 5x10^5 PFU/ml: Phan JC et al. JID, 2016, 214: S222-S228). The novel LFI described in this study could detect Zaire ebolavirus GP (LOD = 1.2 x 10^4 TCID50/ml) or VP40 (9.8 x 10^4 TCID50/ml). Although overall concept is similar to the previous study, this study reports a novel LFI system to detect EBOV VP40 or GP. The manuscript is overall well written, while some more information will likely improve the content.

PLOS authors have the option to publish the peer review history of their article (what does this mean?). If published, this will include your full peer review and any attached files.

Reviewer #1: No

Reviewer #2: No

Reviewer #3: No
---

## [Editor Report · Decision Letter 1]

30 Aug 2020

Dear Dr. AuCoin,

We are pleased to inform you that your manuscript 'Development of an antigen detection assay for early point-of-care diagnosis of Zaire ebolavirus' has been provisionally accepted for publication in PLOS Neglected Tropical Diseases.

Best regards,

Masayuki Saijo

Guest Editor

Samuel Scarpino

Deputy Editor

The authors responded appropriately to the reviewers' comments. This manuscript is well written and revised. I am very glad to inform you that I have decided to make decision, accept, and transfer this manuscript to the production process.

---

## [Editor Report · Acceptance letter]

8 Oct 2020

Dear Dr. AuCoin,

We are delighted to inform you that your manuscript, "Development of an antigen detection assay for early point-of-care diagnosis of *Zaire ebolavirus*," has been formally accepted for publication in PLOS Neglected Tropical Diseases.

Best regards,

Shaden Kamhawi

co-Editor-in-Chief

Paul Brindley

co-Editor-in-Chief
